# FFT-Based Simultaneous Calculations of Very Long Signal Multi-Resolution Spectra for Ultra-Wideband Digital Radio Frequency Receiver and Other Digital Sensor Applications

**DOI:** 10.3390/s24041207

**Published:** 2024-02-13

**Authors:** Chen Wu, Michael Low

**Affiliations:** Defence Research and Development Canada—Ottawa Research Centre, Ottawa, ON K1A 0Z4, Canada; michael.low@forces.gc.ca

**Keywords:** digital receiver, discrete Fourier transform, fast Fourier transform, multi-resolution spectrum, blocking fast Fourier transform, signal processing, field programmable gate array

## Abstract

The discrete Fourier transform (DFT) is the most commonly used signal processing method in modern digital sensor design for signal study and analysis. It is often implemented in hardware, such as a field programmable gate array (FPGA), using the fast Fourier transform (FFT) algorithm. The frequency resolution (i.e., frequency bin size) is determined by the number of time samples used in the DFT, when the digital sensor’s bandwidth is fixed. One can vary the sensitivity of a radio frequency receiver by changing the number of time samples used in the DFT. As the number of samples increases, the frequency bin width decreases, and the digital receiver sensitivity increases. In some applications, it is useful to compute an ensemble of FFT lengths; e.g., $2^{P-j}$ for j=0, 1, 2, …, J, where j is defined as the spectrum level with frequency resolution $\;2^{j}\cdot \Delta f$. Here Δf is the frequency resolution at j=0. However, calculating all of these spectra one by one using the conventional FFT method would be prohibitively time-consuming, even on a modern FPGA. This is especially true for large values of P; e.g., P≥20. The goal of this communication is to introduce a new method that can produce multi-resolution spectrum lines corresponding to sample lengths $\;2^{P-j}$ for all J+1 levels, concurrently, while one long $2^{P}$-length FFT is being calculated. That is, the lower resolution spectra are generated naturally as by-products during the computation of the $2^{P}$-length FFT, so there is no need to perform additional calculations in order to obtain them.

## 1. Introduction

Multi-resolution frequency spectrum analysis is widely used in many areas to process signals from different sources, such as signals from mechanical structures, acoustic sensors, human body signal sensors, and digital RF/microwave receivers. A few examples are given in [1,2,3,4]. Multi-resolution analysis (MRA) is a powerful tool used in image and signal processing. In wavelet analysis [5], a mother wavelet function [6] is scaled in frequency and shifted in time to analyze images or signals at different resolutions [6,7]. Recently, Ref. [8] gives a comprehensive review of the developments, challenges, and opportunities of wavelet analysis, and also covers the evolution of wavelet neural networks. In parallel with the developments in wavelet-based MRA, Fourier transform (FT)-based approaches are also being developed. This framework includes the fractional FT [9,10,11], and the relatively recently developed quadratic-phase FT [12,13], which offers a unified approach to processing transient and non-transient signals and is very useful for analyzing signals with time-varying properties. Although many FT-based MRA methods have been developed, in modern digital receiver development the conventional FT is still widely used. In general, for a fixed digitizer sampling rate, calculating DFTs having different numbers of samples results in spectra having different frequency resolutions. In the FT-based MRA, the short-time Fourier transform (STFT) is often used to calculate signal time-frequency distributions. There are a number STFT-based methods, such as the multi-resolution STFT [14], adaptive STFT [15,16], and STFT frequency-domain (STFT-FD) [17] methods. The idea of the multi-resolution STFT is based on using different window sizes to achieve different frequency resolutions. It uses bandpass filters to divide the signal into frequency bands. Therefore, multi-resolution STFT relies on the use of filters to obtain the multi-resolution spectra. The adaptive STFT adjusts the window size for each time instant based on the local signal characteristics. The STFT-FD computes a transform using the basic idea of the STFT; however, it uses a fixed window size in the frequency domain. This approach is similar to that of the multi-resolution STFT and wavelet methods, but with a different methodology. The key of the STFT-FD is to define the window size as the number of cycles of the frequency component being considered rather than defining it in time. The STFT-FD needs neither filtering nor the evaluation of local signal characteristics as in the multi-resolution STFT and the adaptive STFT, respectively.

It is well known that any STFT-based time-to-frequency transform has a limitation. This is the inherent trade-off between time and frequency resolution, i.e., the short-time window has good time resolution, but poor frequency resolution, and vice versa. In [18], the piecewise-constant window blocking fast Fourier transform (PCW-BFFT) method was introduced for ultra-wideband digital receivers in order to achieve a fine frequency resolution and a high receiver sensitivity. By applying a number of window functions with different widths in the time domain, the PCW-BFFT method can simultaneously produce multi-time-resolution spectrograms with a very fine frequency resolution for a very long-time sequence (e.g., $N=2^{20}$); however, this method cannot produce multi-frequency-resolution spectra concurrently.

The DFT-based time-to-frequency transform is the most widely used signal processing method in ultra-wideband digital sensors and is commonly implemented using the fast Fourier transform (FFT) on field programmable gate arrays (FPGAs) in real-time applications. Ref. [19] presents several methods using the FFT to detect and analyze intercepted signals with multi-resolution frequency spectral lines (MR-FSL). However, up to now, to obtain the MR-FSL of a signal, multiple FFT calculations with different signal lengths (e.g., $N=2^{P-j}$ where j=0, 1, 2, …, J) must be performed one by one. The disadvantages of the traditional FFT-based methods are: (1) for large signal length (e.g., P=20 and j=0, 1, 2, 3), four different long-length FFTs have to be calculated; (2) one has to wait for a large number of time samples to be generated by the analog-to-digital converter (ADC) in the receiver before the FFTs can be performed; (3) many FPGA resources are needed to store those time samples before the FFTs can be performed; and (4) more importantly, the computational effort spent on the calculation of the longest FFT cannot be reused in the calculations of any of the shorter length FFTs.

To mitigate the problem of requiring a large amount of hardware resources to compute very long length FFTs and effectively calculate the frequency spectrum of long-sequence signals in an embedded device in real time, the blocking FFT (BFFT) method was first introduced in [20]. It also demonstrated the implementation of the method on Xilinx’s ZYNQ-7000 device. In order to perform an  N-point frequency bin calculation, the BFFT conceptually divides  N  time samples into  K  consecutive timeslot blocks consisting of  M samples each. As each block of $\;M\;\left(=N/K\right)\;$ samples is generated by the ADC, the BFFT method performs K  independent  M-point FFTs, where suitable phase rotation factors are applied to the  M  input samples of each of the K  FFTs. Each of these FFTs can be computed in parallel (e.g., on an FPGA or GPU). The resulting K×M=N  frequency data are accumulated with the results from previous timeslots to form an N-bin spectrum line. Each spectrum line can be viewed as a group of K equal-length frequency blocks where the $k^{th}$ frequency block has frequency bins $f_{kq}$, where k=0, 1, 2, …, K−1,  and $q=k,\;M+k,\;\ldots \;,\left(K-1\right)M+k$. The final N-bin spectrum line is calculated as the accumulation of the frequency results from processing all K  timeslots. The merit of the BFFT method is that instead of having to first store a large number of  N  time samples before being able to compute a single, long N-point FFT, which may be a challenge even on today’s high-end FPGAs, the BFFT performs K many M-point FFTs in parallel, as soon as M  samples are available from the current timeslot. This tremendously reduces the hardware resource requirements, produces useful intermediate results with low latency, and also enables the development of other useful methods for ultra-wideband digital receiver designs, such as the multi-time-resolution piecewise-constant window blocking Fourier transform (PCW-BFFT) method [18], the accumulatively increasing receiver sensitivity (AIRS) method [21], and the multi-frequency-resolution spectrum line calculation method introduced in this paper.

Taking advantage of the aforementioned BFFT process, this paper for the first time introduces an FFT-based MR-FSL calculation method, called the multi-frequency-resolution BFFT (MFR-BFFT). It can calculate the spectrum line having $N_{0}=2^{P}\;$ frequency bins, while simultaneously producing all the spectrum lines having $N_{j}=2^{P-j}$ frequency bins, where j=0,1, 2, …, J, without needing to perform extra FFT calculations on the M samples from the current timeslot. That is, the computational efforts to obtain the $2^{P}$-bin spectrum line are reused to obtain the spectrum lines that have $2^{P-1},\;\;2^{P-2},\;\;2^{P-3},\;\ldots ,\;2^{P-J}\;$ frequency bins at other frequency resolution levels. With a fixed ADC sampling rate (i.e., $f_{s}=\frac{1}{t_{s}}$ where $t_{s}$ is the time interval between samples), the MFR-BFFT uses $2^{P-j}\;$ time samples (where j=0, 1, 2,…, J) to compute spectrum lines up to $J=\log_{2}\left(K_{0}\right)$ levels with $2^{j}$ different spectrum lines at Level-*j*. Here, $K_{0}$ is the number of timeslots in Level-0. Since the ADC rate is fixed, the frequency resolutions at different spectrum levels are $\;2^{j}\cdot \Delta f$, where $\Delta f=f_{s}/2^{P}$ is the Level-0 frequency resolution. The signal duration used to get different level spectrum lines is $N_{j}\times t_{s}$. More details will be presented in the next section.

Compared with other aforementioned existing MRA methods, although the MFR-BFFT method does not have the flexibility to obtain different levels of resolutions as wavelet-based MRA methods do, it can tremendously reduce the computational effort on hardware to produce spectra with a fixed set of frequency resolutions. Compared with other FT-based MRA methods, the MFR-BFFT has the advantage of being able to process very long-length signal samples, and to produce a set of different frequency resolution spectra simultaneously. Note that the very long-length signal samples do not necessarily mean a long signal in the time domain. It can be a long signal sampled at a low ADC sampling rate, or a very short signal sampled at high sampling rate. For example, if an ADC samples at 2.5 GSample/s, within about 0.42 ms, it can produce $2^{20}\;$ time samples.

The rest of the paper is organized as follows: The derivations and discussions of the MFR-BFFT methods are presented in the next section. Section 3 shows the MFR-BFFT results using data obtained from an ultra-wideband digital radio receiver and compares the results with those obtained from the “fft” function of Mathworks MATLAB (R2023b) to prove the correctness of the MFR-BFFT implementation in MATLAB. The conclusion can be found in the last section. The hardware implementation including an expansion of the comparative analysis between MFR-BFFT and conventional FFT, such as computational complexity, power consumption, and execution time, etc., will be reported in another full paper.

## 2. FFT-Based Multi-Frequency-Resolution Spectrum Line Calculation

Based on the discrete Fourier transform (DFT), the frequency spectrum line at Level-0 can be expressed as: (1)$$X_{01}\left(v_{0}\right)=\overset{N_{0}-1}{\underset{n=0}{\sum }}\;x\left(n\right)W_{N_{0}}^{nv_{0}}\;\;$$
where $\;v_{0}=0,\;1,\;2,\;\ldots ,\;N_{0}-1$ are the frequency bin centers on the Level-0 spectrum line, $W_{N_{0}}^{nv_{0}}=e^{-\frac{\;2\sqrt{-1}\pi nv_{0}}{N_{0}}}$, and x  is a complex-valued signal. $X_{01}\;$ means the first spectrum line at Level-0, although at this level there is only one spectrum line. Using the BFFT, $N_{0}$ time samples are divided into $K_{0}$ equal timeslots and each timeslot has M samples; therefore, (1) can be written as:(2)$$X_{01}\left(v_{0}\right)=\overset{K_{0}-1}{\underset{k=0}{\sum }}\left\{\overset{\left(k+1\right)M-1}{\underset{n=kM}{\sum }}x\left(n\right)W_{N_{0}}^{nv_{0}}\right\}$$

By substituting  u=n−kM, then for the $k^{th}$ timeslot of M samples, the portion inside the curly braces can be rewritten as:(3)$$\overset{\left(k+1\right)M-1}{\underset{n=kM}{\sum }}x\left(n\right)W_{N_{0}}^{nv_{0}}=\overset{M-1}{\underset{u=0}{\sum }}x\left(u+kM\right)W_{N_{0}}^{\left(u+kM\right)v_{0}}$$

We now define:(4)$$Y_{01_{k}}\left(v_{0}\right)=\overset{M-1}{\underset{u=0}{\sum }}x\left(u+kM\right)W_{N_{0}}^{\left(u+kM\right)v_{0}}$$
which is the frequency information obtained from the $k^{th}$ timeslot of M samples for the Level-0 spectrum line. Using (4), (2) can be rewritten as:(5)$$X_{01}\left(v_{0}\right)=\overset{K_{0}-1}{\underset{k=0}{\sum }}Y_{01_{k}}\left(v_{0}\right)$$

In order for the M samples in the $k^{th}$ timeslot to contribute to the overall $N_{0}$ bins, the BFFT also divides $N_{0}$ frequency bins into $K_{0}$ equal-length frequency blocks, where each frequency block also has M  frequency bins. By substituting $\;v_{0}=m_{0}+rK_{0}$, where $m_{0}=0,\;1,\;2,\;\ldots ,\;K_{0}-1$ to define the total $K_{0}\;$ frequency blocks at Level-0, and  r=0, 1, 2, …,M−1 to specify the M frequency bins in each frequency block, then (4) becomes:(6)$$Y_{01_{k}}\left(m_{0}+rK_{0}\right)=\overset{M-1}{\underset{u=0}{\sum }}x\left(u+kM\right)W_{N_{0}}^{m_{0}\left(u+kM\right)}W_{M}^{ru}$$
where $MK_{0}=N_{0}$ and $W_{N_{0}}^{kMK_{0}r}=1\;$ are used. Hence, using the BFFT to calculate (1), we have:(7)$$X_{01}\left(m_{0}+rK_{0}\right)=\overset{K_{0}-1}{\underset{k=0}{\sum }}\left\{\overset{M-1}{\underset{u=0}{\sum }}x\left(u+kM\right)W_{N_{0}}^{m_{0}\left(u+kM\right)}W_{M}^{ru}\right\}$$

It can be seen that to contribute to all the frequency bins in $X_{01}$ using M samples in the $k^{th}$ timeslot, the samples need to be multiplied by phase rotation factors $W_{N_{0}}^{m_{0}\left(u+kM\right)}$, which are determined by the frequency block ($m_{0})$ and time index $\left(n=u+kM\right)$ in the overall time sequence, before performing the M-point FFT (the operations specified in curly braces) indicated in (7). Since $m_{0}=,\;1,\;2,\;\ldots ,\;K_{0}-1$, a total of $K_{0}$ many M-point FFTs are computed at Level-0.

Now, let us consider the Level-1 spectrum line calculations with the same time sequence $x\left(n\right)$ where $\left(n=0,\;1,\;2,\;\ldots ,\;N_{0}-1\right)$. Using the DFT, the spectrum lines can be computed as:(8)$$X_{1i}\left(v_{1}\right)=\overset{iN_{1}-1}{\underset{n=\left(i-1\right)N_{1}}{\sum }}\;x\left(n\right)W_{N_{1}}^{nv_{1}}\;\;$$
where i=1, 2 indicates that there are the two spectrum lines in Level-1. The first is calculated from samples *0* to $N_{1}-1$, and the second is calculated from samples $N_{1}\;$ to $\;N_{0}-1$. Here $N_{1}=N_{0}/2$ and $\;W_{N_{1}}^{nv_{1}}=e^{-\frac{\;2\sqrt{-1}\pi nv_{1}}{N_{1}}}$. Note that (1) these two spectrum lines have the same frequency bins $v_{1}=0,\;1,\;2,\;\ldots ,\;N_{1}-1$; (2) they only use half of the available samples in $x\left(n\right)$ compared with what was used in Level-0; and (3) the frequency bin size at Level-1 is double that of Level-0, since the ADC sampling rate is fixed.

Again, if we use the BFFT to calculate the Level-1 spectrum lines with the same number of samples $\left(M\right)$ in a timeslot that is used at Level-0, we have:(9)$$X_{1i}\left(v_{1}\right)=\overset{iK_{1}-1}{\underset{k=\left(i-1\right)K_{1}}{\sum }}\left\{\overset{\left(k+1\right)M-1}{\underset{n=kM}{\sum }}x\left(n\right)W_{N_{1}}^{nv_{1}}\right\}$$
where the first and second $K_{1}\left(=K_{0}/2\right)\;$ groups of timeslots are used to calculate $X_{11}\left(v_{1}\right)$ and $X_{12}\left(v_{1}\right)$, respectively. By substituting  u=n−kM, then for the $k^{th}$ timeslot’s M samples, we have:(10)$$\overset{\left(k+1\right)M-1}{\underset{n=kM}{\sum }}x\left(n\right)W_{N_{1}}^{nv_{1}}=\overset{M-1}{\underset{u=0}{\sum }}x\left(u+kM\right)W_{N_{1}}^{\left(u+kM\right)v_{1}}$$

In order for the M samples in $k^{th}$ timeslot to contribute to the overall $N_{1}$ bins, the BFFT also divides the $N_{1}$ frequency bins into $K_{1}$ blocks, each of which contains M  frequency bins. By substituting $\;v_{1}=m_{1}+rK_{1}$, where $m_{1}=0,\;1,\;2,\;\ldots ,\;K_{1}-1$ to define $K_{1}\;$ frequency blocks in each spectrum line, and  r=0, 1, 2, …,M−1 (i.e., there are M frequency bins in each frequency block), then from (10) we have:(11)$$Y_{1_{k}}\left(m_{1}+rK_{1}\right)=\overset{M-1}{\underset{u=0}{\sum }}x\left(u+kM\right)W_{N_{1}}^{m_{1}\left(u+kM\right)}W_{M}^{ru}$$
where $MK_{1}=N_{1}$ and $W_{N_{1}}^{kMK_{1}r}=1$ are used. Then, using the BFFT, the spectrum lines at Level-1 can be expressed as:(12)$$X_{1i}\left(m_{1}+rK_{1}\right)=\overset{iK_{1}-1}{\underset{k=\left(i-1\right)K_{1}}{\sum }}\left\{\overset{M-1}{\underset{u=0}{\sum }}x\left(u+kM\right)W_{N_{1}}^{m_{1}\left(u+kM\right)}W_{M}^{ru}\right\}$$

In general, using the BFFT, the spectrum lines at Level-*j* can be expressed as:(13)$$X_{ji}\left(m_{j}+rK_{j}\right)=\overset{iK_{j}-1}{\underset{k=\left(i-1\right)K_{j}}{\sum }}\left\{\overset{M-1}{\underset{u=0}{\sum }}x\left(u+kM\right)W_{N_{j}}^{m_{j}\left(u+kM\right)}W_{M}^{ru}\right\}$$
where $j=0,\;1,\;2,\;\ldots ,\;\log_{2}(K_{0})$ defines the spectrum line levels; $i=1,\;2,\;3,\;\ldots ,2^{j}$ denotes the spectrum lines in Level-*j*; and the $i^{th}$ spectrum line is obtained using timeslots from $\left(i-1\right)K_{j}\;$ to $iK_{j}-1,$ where $K_{j}\;$ specifies the number of timeslots in Level-*j*. 

From the above discussions we can observe that: 1.with fixed $N_{0}\;$ samples used in Level-0, the maximum number of levels is limited by the number of samples $\left(M\right)$ in a timeslot to $J=\log_{2}(K_{0})=\log_{2}(N_{0}/M)$;2.although the M samples of a given timeslot can be used in all levels, the differences are: the number of M-point FFTs is determined by $\;K_{j}\;\left(=K_{0}/2^{j}\right)$, andthe number of frequency bins is determined by $\;N_{j}\;\left(=N_{0}/2^{j}\right)$;3.more importantly, at different levels, the same M samples in the $k^{th}$ timeslot are multiplied with phase rotation factors $W_{N_{j}}^{m_{j}\left(u+kM\right)}$ in (13), which can be rewritten as $W_{N_{0}}^{2^{j}\cdot m_{j}\left(u+kM\right)}$, given that $N_{j}=N_{0}/2^{j}$.

Therefore, (13) can be expressed as:(14)$$X_{ji}\left(m_{j}+rK_{j}\right)=\overset{iK_{j}-1}{\underset{k=\left(i-1\right)K_{j}}{\sum }}\left\{\overset{M-1}{\underset{u=0}{\sum }}x\left(u+kM\right)W_{N_{0}}^{2^{j}\cdot m_{j}\left(u+kM\right)}W_{M}^{ru}\right\}$$

Comparing the portion inside the curly braces in (14) with the portion inside the curly braces in (7), we find that after we calculate the DFT (the portion inside the curly braces of (7)) using M samples in a timeslot, the DFT results can not only be used for frequency bins at Level-0, but some of them can also be used for other-level spectrum lines, as long as: 1.these other-level spectrum lines also use M samples from the current timeslot for the calculation; and2.the frequency block number $m_{j}$ satisfies the relation given in (15), which means these other-level spectrum lines not only share the timeslot, but also use the same phase rotations factors.

(15)$$2^{j}\cdot m_{j}=m_{0}\;$$ where again $m_{0}=0,\;1,\;2,\;\ldots ,\;K_{0}-1$, and $m_{j}=0,\;1,\;2,\;\ldots ,\;K_{j}-1$, where $j=0,\;1,\;2,\;\ldots ,\;\log_{2}(K_{0})$ and $K_{j}=K_{0}/2^{j}$. 

The above discussion infers that the other-level spectrum lines can be obtained at the same time that the Level-0 spectrum line is being calculated using the BFFT. Since the ADC sampling rate is fixed, and the number of time samples is reduced by half at each progressively higher spectrum level, while the frequency resolution is also reduced by half. The number of spectrum lines is doubled when going from Level-*j* to Level-*(j + 1)* if all the time samples used in Level-0 are used in the computation of the spectrum lines at other levels.

To make the above discussion clear, let us see an example. Here, we use $N_{0}=32$ time samples with M=4 samples in each timeslot. Therefore, there are $K_{0}=8$ timeslots, and there are 4 levels of spectrum lines. Table 1 shows the timeslots and time samples that are shared between spectrum lines ($X_{ji})$ at different levels. For example, 4 samples in TS=0 are used in the calculations of $X_{01}$, $X_{11}$, $X_{21}$, and $X_{31}$, and 4 samples in TS=4 are used in the calculations of $X_{01}$, $X_{12}$, $X_{23}$, and $X_{35}$.

There are 8, 4, 2, and 1 frequency blocks at Level-0, -1, -2, and -3, respectively. Table 2 shows that the Level-0 *M*-point FFT results can also be used to generate other-level spectrum lines, as long as the samples in a timeslot

are shared between these spectrum lines (see Table 1), andare multiplied by the same phase rotation factors that meet the criterion described in (15).

## 3. Comparisons of MFR-BFFT Results with FFT Results

Figure 1 shows a segment of I/Q data recorded by an ultra-wideband digital receiver. There is a total of $2^{20}$ samples at Level-0, and M=256 samples in each timeslot. Four spectrum levels are calculated with $\;2^{20}$, $2^{19}$, $2^{18}$, and $2^{17}$ frequency bins at each of the different levels. The ADC sampling rate is 533.333 MHz, so that the frequency resolutions are 508.6, 1017.3, 2034.5, and 4069.0 Hz for Level-0 through Level-3, respectively.

Figure 2 shows the MFR-BFFT results at different spectrum levels. To verify the MFR-BFFT method, the results obtained using the MATLAB “fft” function are overlaid in the figure (with an arbitrary offset in order to improve visibility). The error distributions between the results in Figure 2 are shown in Figure 3 using histogram plots. The very small errors, which are due to differences in the order in which numerical rounding has occurred, show the correctness of our MFR-BFFT method.

## 4. Conclusions

This communication introduces a new multi-resolution spectrum line calculation method, called the multi-frequency-resolution blocking fast Fourier transform (MFR-BFFT). It can simultaneously calculate a set of very long multi-resolution signal spectra by reusing the BFFT results on all samples used to calculate the finest frequency resolution spectrum. The paper gives the detailed derivation of the MFR-BFFT method and uses actual recorded data to demonstrate the use of the method. The MFR-BFFT results are also compared with conventional FFT results. They are identical for all practical purposes. The success of the MFR-BFFT in simultaneously calculating multi-frequency-resolution spectrum lines using the same set of samples leads to a new and efficient approach for analyzing long-length digitized samples in real-time systems, such as FPGA-based hardware. Although this paper discusses the MFR-BFFT method for applications with long-length signals, this method can be considered for any applications needing multi-resolution spectra and using FPGAs for near-real-time calculations. In follow-on work, the MFR-BFFT method will be combined with the PCW-BFFT method to develop a new DFT-based time-frequency distribution calculation method that can concurrently produce multi-frequency-resolution and multi-time-resolution spectrograms, while consuming digitized data in K smaller blocks of M samples produced by high-speed ADCs.

## Figures and Tables

**Figure 1 sensors-24-01207-f001:**
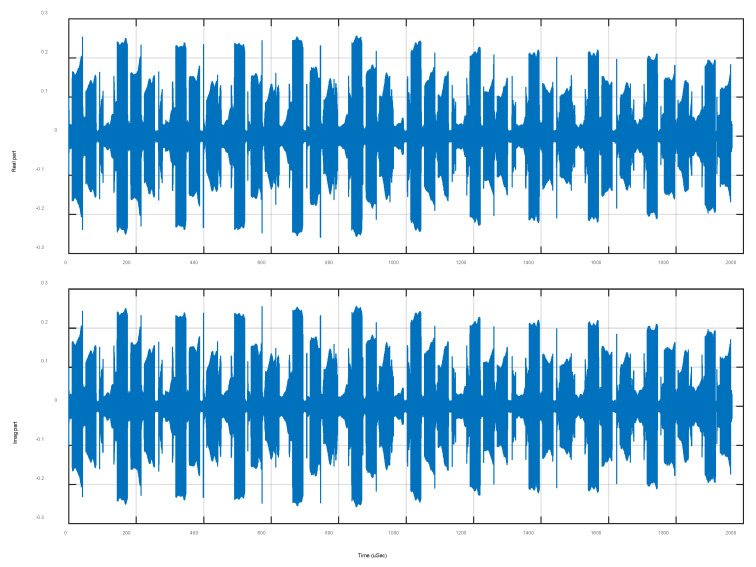
A segment of a digitized I/Q signal with a total of $2^{20}$ samples.

**Figure 2 sensors-24-01207-f002:**
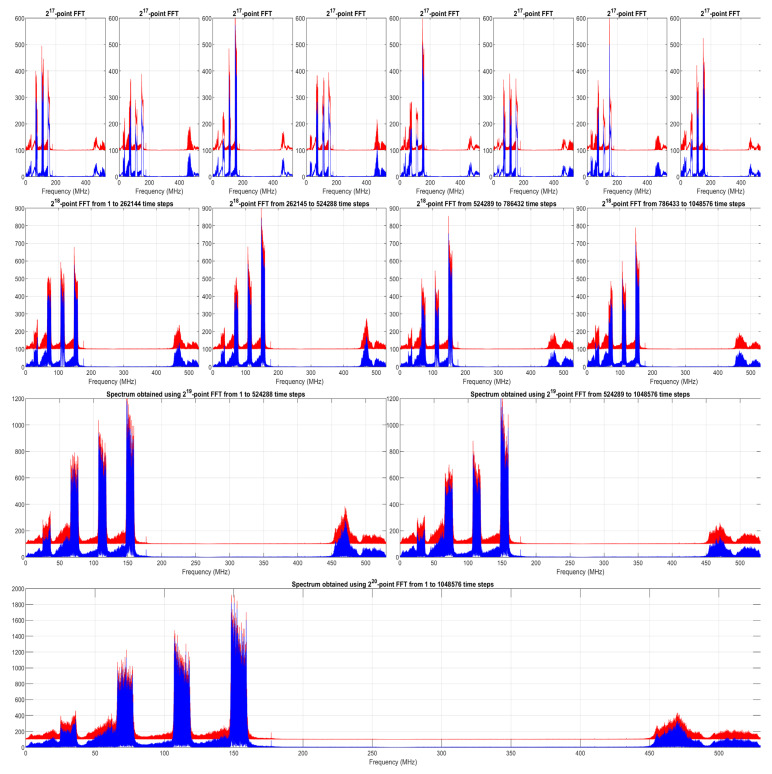
Spectrum lines calculated using different numbers of samples using the MFR-BFFT (blue) and the MATLAB “fft” function (red). Since results are very close, an arbitrary offset of 100 was added to the magnitude of the MATLAB results in order to improve visibility.

**Figure 3 sensors-24-01207-f003:**
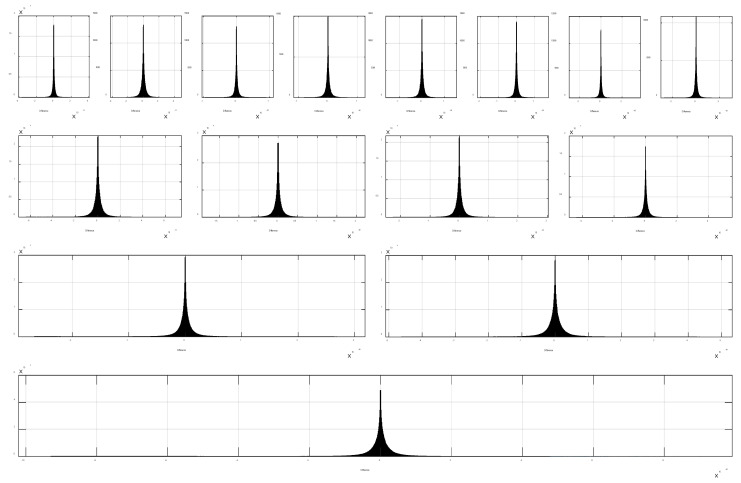
Histograms of the differences between the results in Figure 2 obtained using the MFR-BFFT method and the MATLAB “fft” function at different frequency resolution levels.

**Table 1 sensors-24-01207-t001:** Timeslots (TSs) and time samples used in different levels spectrum line (SL) calculations.

TS	0				1				2				3				4				5				6				7			
**SL**	**Time Samples** $x\left(n\right)$																															
$X_{01}$	0	1	2	3	4	5	6	7	8	9	10	11	12	13	14	15	16	17	18	19	20	21	22	23	24	25	26	27	28	29	30	31
$X_{11}$	0	1	2	3	4	5	6	7	8	9	10	11	12	13	14	15																
$X_{12}$																	16	17	18	19	20	21	22	23	24	25	26	27	28	29	30	31
$X_{21}$	0	1	2	3	4	5	6	7																								
$X_{22}$									8	9	10	11	12	13	14	15																
$X_{23}$																	16	17	18	19	20	21	22	23								
$X_{24}$																									24	25	26	27	28	29	30	31
$X_{31}$	0	1	2	3																												
$X_{32}$					4	5	6	7																								
$X_{33}$									8	9	10	11																				
$X_{34}$													12	13	14	15																
$X_{35}$																	16	17	18	19												
$X_{36}$																					20	21	22	23								
$X_{37}$																									24	25	26	27				
$X_{38}$																													28	29	30	31

**Table 2 sensors-24-01207-t002:** The  M-point FFT results can be reused at different levels $\;\left(K_{0},\;K_{1},\;K_{2},\;K_{3}\right)=\left(8,\;4,\;2,\;1\right)$.

$X_{01}$	$X_{1i}$	$X_{2i}$	$X_{3i}$
$\overset{K_{0}-1}{\underset{k=0}{\sum }}\left\{\overset{M-1}{\underset{u=0}{\sum }}x\left(u+kM\right)W_{N_{0}}^{0\left(u+kM\right)}W_{M}^{ru}\right\}$	$\overset{iK_{1}-1}{\underset{k=\left(i-1\right)K_{1}}{\sum }}\left\{\overset{M-1}{\underset{u=0}{\sum }}x\left(u+kM\right)W_{N_{0}}^{0\left(u+kM\right)}W_{M}^{ru}\right\}$	$\overset{iK_{2}-1}{\underset{k=\left(i-1\right)K_{2}}{\sum }}\left\{\overset{M-1}{\underset{u=0}{\sum }}x\left(u+kM\right)W_{N_{0}}^{0\left(u+kM\right)}W_{M}^{ru}\right\}$	$\overset{iK_{3}-1}{\underset{k=\left(i-1\right)K_{3}}{\sum }}\left\{\overset{M-1}{\underset{u=0}{\sum }}x\left(u+kM\right)W_{N_{0}}^{0\left(u+kM\right)}W_{M}^{ru}\right\}$
$\overset{K_{0}-1}{\underset{k=0}{\sum }}\left\{\overset{M-1}{\underset{u=0}{\sum }}x\left(u+kM\right)W_{N_{0}}^{1\left(u+kM\right)}W_{M}^{ru}\right\}$			
$\overset{K_{0}-1}{\underset{k=0}{\sum }}\left\{\overset{M-1}{\underset{u=0}{\sum }}x\left(u+kM\right)W_{N_{0}}^{2\left(u+kM\right)}W_{M}^{ru}\right\}$	$\overset{iK_{1}-1}{\underset{k=\left(i-1\right)K_{1}}{\sum }}\left\{\overset{M-1}{\underset{u=0}{\sum }}x\left(u+kM\right)W_{N_{0}}^{2\left(u+kM\right)}W_{M}^{ru}\right\}$		
$\overset{K_{0}-1}{\underset{k=0}{\sum }}\left\{\overset{M-1}{\underset{u=0}{\sum }}x\left(u+kM\right)W_{N_{0}}^{3\left(u+kM\right)}W_{M}^{ru}\right\}$			
$\overset{K_{0}-1}{\underset{k=0}{\sum }}\left\{\overset{M-1}{\underset{u=0}{\sum }}x\left(u+kM\right)W_{N_{0}}^{4\left(u+kM\right)}W_{M}^{ru}\right\}$	$\overset{iK_{1}-1}{\underset{k=\left(i-1\right)K_{1}}{\sum }}\left\{\overset{M-1}{\underset{u=0}{\sum }}x\left(u+kM\right)W_{N_{0}}^{4\left(u+kM\right)}W_{M}^{ru}\right\}$	$\overset{iK_{2}-1}{\underset{k=\left(i-1\right)K_{2}}{\sum }}\left\{\overset{M-1}{\underset{u=0}{\sum }}x\left(u+kM\right)W_{N_{0}}^{4\left(u+kM\right)}W_{M}^{ru}\right\}$	
$\overset{K_{0}-1}{\underset{k=0}{\sum }}\left\{\overset{M-1}{\underset{u=0}{\sum }}x\left(u+kM\right)W_{N_{0}}^{5\left(u+kM\right)}W_{M}^{ru}\right\}$			
$\overset{K_{0}-1}{\underset{k=0}{\sum }}\left\{\overset{M-1}{\underset{u=0}{\sum }}x\left(u+kM\right)W_{N_{0}}^{6\left(u+kM\right)}W_{M}^{ru}\right\}$	$\overset{iK_{1}-1}{\underset{k=\left(i-1\right)K_{1}}{\sum }}\left\{\overset{M-1}{\underset{u=0}{\sum }}x\left(u+kM\right)W_{N_{0}}^{6\left(u+kM\right)}W_{M}^{ru}\right\}$		
$\overset{K_{0}-1}{\underset{k=0}{\sum }}\left\{\overset{M-1}{\underset{u=0}{\sum }}x\left(u+kM\right)W_{N_{0}}^{7\left(u+kM\right)}W_{M}^{ru}\right\}$			

## Data Availability

All simulated data are available to the readers.

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
