# Peer review of "FFT-Based Simultaneous Calculations of Very Long Signal Multi-Resolution Spectra for Ultra-Wideband Digital Radio Frequency Receiver and Other Digital Sensor Applications"

_sensors, 2024, doi:10.3390/s24041207_

Round 1
Reviewer 1 Report
Comments and Suggestions for Authors
The paper well drafted and organized. I recommend publish the paper with the following major revisions
The authors have reviewed very less number of references, it is advised authors enhance the introduction section by adding additional references.
There by you can also add the recent references to compare the proposed work to evaluate the originality and performance of your proposed work.
There are many FFT based Digital Radio and sensing systems available in the open literature, So it is advised authors form a comparison table with existing literature for the performance parameters.
Author Response
Dear Reviewer
Please see attached file.
Many thanks
Chen

Reviewer 2 Report
Comments and Suggestions for Authors
Overview and strength:
1. The paper addresses a significant issue in the field of digital sensor design, proposing a new method for simultaneous calculation of multi-resolution spectra.
2. The use of the fast Fourier transform (FFT) algorithm and its application in hardware, such as field programmable gate arrays (FPGAs), demonstrates the practical relevance of the research.
3. The proposed method has the potential to significantly reduce the computational resources required, especially for very large values of P, which is a valuable contribution to the field.
Weaknesses and requested revisions:
1. The abstract could be more concise and clearer in explaining the novelty and significance of the proposed method.
2. The paper lacks a detailed discussion of the potential limitations or challenges associated with the new method, which would provide a more comprehensive understanding of its applicability.
3. While the paper introduces a new method, it would benefit from a more in-depth comparison with existing approaches to highlight its advantages and limitations more effectively.
4. Providing a more detailed explanation of the specific scenarios or applications in which the proposed method would be most beneficial could help readers understand its practical significance more effectively.
5. Including a discussion on the potential trade-offs or drawbacks of the new method, such as any limitations in accuracy or specific conditions under which it may not be suitable, would contribute to a more balanced evaluation of its utility.
6. Incorporating a comparison with alternative methods, along with a discussion of how the proposed method addresses their limitations or offers unique advantages, would strengthen the paper's contribution to the field.
By addressing these points, the paper can further enhance its clarity, impact, and practical relevance, ultimately strengthening its contribution to the field of digital sensor design and signal processing.
Author Response

(The authors gave the same response as above.)

Reviewer 3 Report
Comments and Suggestions for Authors
This paper introduces the multi-frequency-resolution blocking fast Fourier transform (MFR-BFFT) method, which enables efficient, simultaneous calculation of multi-resolution signal spectra on FPGA platforms, offering a novel approach for real-time analysis of long-length digitized samples. Due to the interest of the topic that it addresses, I find the work of utility for the scientific community. In this sense, I think that it could be suitable for publication in the Diagnostics journal provided that the following comments are implemented within the document:
- The introduction could benefit from a more detailed explanation of the problem under study and the specific contribution of the proposed method when compared with previously published papers.
- Discuss any optimization techniques used in implementing the MFR-BFFT on FPGA platforms, considering FPGA resource constraints.
- An expansion of the comparative analysis between MFR-BFFT and conventional FFT would be welcome, including more metrics for comparison, such as computational complexity, power consumption, and execution time.
- Provide more examples of real-world scenarios where MFR-BFFT could be particularly beneficial. Moreover, discuss potential applications in various fields like telecommunications, audio processing, or biomedical engineering. Finally, explore how this method connects with or can be applied in other fields, like machine learning or quantum computing.
- Outline potential enhancements to the MFR-BFFT method. This might include integration with other signal processing techniques or adaptations for specific types of signal analysis.
- Discuss the robustness of the method against various signal types and its limitations.
Comments on the Quality of English Language-
Author Response

(The authors gave the same response as above.)

Round 2
Reviewer 1 Report
Comments and Suggestions for Authors
The paper is well organized and modified. Addressed all the issues that were raised in the first review.
I recommend to accept the publication in the present form.
Reviewer 2 Report
Comments and Suggestions for Authors
Answer of authors are accepted.